# Comparison of Two Different Approaches to Treat a Hallux Valgus Deformity: Intramedullary Self-Locked Plates and Herbert Screws

**DOI:** 10.3390/medicina55090595

**Published:** 2019-09-16

**Authors:** Zekeriya Okan Karaduman, Ozan Turhal, Yalçın Turhan, Mehmet Arıcan, Cemal Güler, Sengul Cangur

**Affiliations:** 1Department of Orthopaedics and Traumatology, Medical Faculty, Duzce University, Duzce 81000, Turkey; yturhan_2000@yahoo.com (Y.T.); ari_can_mehmet@hotmail.com (M.A.); 2Department of Orthopaedics and Traumatology, Duzce State Hospital, Duzce 81000, Turkey; 3Department of Orthopaedics and Traumatology, Medical Faculty, Çorum Hitit University, Çorum 19000, Turkey; drcemal78@gmail.com; 4Department of Biostatistics, Medical Faculty, Duzce University, Duzce 81000, Turkey; sengulcangur@duzce.edu.tr

**Keywords:** hallux valgus, distal metatarsals osteotomy, intramedullary self-locking plate, Herbert screw

## Abstract

*Background and objectives:* Hallux valgus is a complex deformity of the first metatarsophalangeal joint characterized by varus deformity of the first metatarsal bone, valgus deformity of the big toe, and lateral deviation of the extensor tendons and sesamoid bones. Several surgical methods have been described for correction of the deformity. Different materials have been used for the fixation of osteotomy. We compared the functional, radiological, and pain results of intramedullary self-locked plates and Herbert screws for the treatment of a hallux valgus deformity. *Materials and Methods:* Distal metatarsals were treated with self-locking intramedullary plate–screw systems in 18 feet from 12 patients (Group 1) and with Herbert screws in 18 feet from 12 patients (Group 2). The hallux valgus angle (HVA) and intermetatarsal angle (IMA) in patients of Group 1 and 2 were examined radiologically during the pre- and postoperative periods. We also determined the American Orthopedic Foot and Ankle Society (AOFAS), EQ-5D General Life Quality Scale, and Visual Analogue Scale (VAS) scores during the pre- and postoperative periods and compared the scores between groups. *Results:* Post hoc test results of HVA and IMA angles measured after the operation were significantly higher in Group 2 than in Group 1. In each group, the AOFAS scores during the preoperation period were significantly lower than those during the postoperation period (*p* < 0.001). According to the post hoc test results, the VAS scores after the operation were significantly higher in Group 2 than in Group 1 (*p* < 0.001). *Conclusions:* For the surgical treatment of hallux valgus, operations using self-locked plates compared to a single screw are superior in terms of providing rigid stability and for functional, radiological, and pain scores.

## 1. Introduction

Hallux valgus is a deformity of the front area of the foot characterized by a deformity of the first metatarsophalangeal knuckle showing progressive valgus and lateral deformities. Because it is a progressive deformity, it often results in secondary osteoarthritis and hammer toe [1,2,3]. Hallux valgus (HV) was first defined by the German surgeon Carl Hueter in 1870 as an abduction-valgus deformity of the toe [1,2]. According to a recent report, its rate among individuals between the ages of 18 and 65 is 23%, among those over age 65 the rate is 35.7%, and among adolescents it is 7.8% [1]. Hallux valgus is a common condition that appears to be strongly associated with age and female sex [2]. 

Among the surgical treatments, there are more than 150 options including bunionectomy, metatarsals, phalangeal osteotomy, resection, interposition arthroplasty, and metatarsophalangeal joint arthrodesis [3,4]. Clinically, the primary objective is to ensure subject satisfaction, to correct the valgus deformity cosmetically, and to relieve the ache in the first MTP joint. In radiological terms, the valgus angle before the operation is the primary criteria of valgus healing [5,6]. In this study, we compared clinical and radiological results of the two treatment options (intramedullary self-locked plates and Herbert screws) of hallux valgus patients after distal metatarsal osteotomy.

## 2. Materials and Methods

### 2.1. Participants

All patients provided informed consent, the study design was approved by the Düzce University Clinical Research Ethics Committee (Düzce, Turkey) (no. 2018/171), and the study was performed in accordance with the principles of the Declaration of Helsinki. This retrospective study includes patients with high to moderate HV, who were above 18 years old, and underwent distal metatarsal osteotomy with the Lindgren–Turan technique between the years 2013 and 2016. A total of 24 patients who were surgically treated were included in this study. Patients diagnosed with hallux valgus with an ache in the first MTP joint, with trouble putting shoes on, with cosmetic complaints, or with sufficient joint range of motion and radiologically over 20 degrees of hallux valgus angle (HVA) and over 10 degrees of intermetatarsal angle (IMA) were included in the study [7] (Figure 1). 

Patients with results indicative of osteoarthritis and gout and those who could not adhere to routine clinical follow-ups were not included. Patients in Group 1 to whom the self-locked intramedullary platter was administered were observed for 5 (3–12) months, while patients in Group 2 to whom the Herbert screw was administered were observed for 14 months (8–30). During the postoperative period, short leg splints were administered only to patients in Group 2 for 4 weeks. A total of 18 feet from 12 patients were fixed using an intramedullary platter and another 18 feet from 12 patients were treated using Herbert screw (Figure 2).

### 2.2. Surgical Technique

The surgical procedures were performed in the supine position under spinal anesthesia and with a pneumatic tourniquet by the same surgical team. An approximately 3 cm dorsal medial incision over the first MTP joint and a linear capsular incision were made. Release of articular space and of soft tissues was made from same incision. The bunion was then shaved (exostectomy). Osteotomy of Lindgren–Turan was administered to all patients.

### 2.3. Radiological and Clinical Evaluation

All patients were radiologically evaluated during both the pre- and postoperative period in terms of HVA and IMA. During radiographic imaging, the tube is placed 100 cm away from the foot and focusing on the midtarsal joint. The tube is than rotated at a 15° angle to the ankle relative to the plantar face of the foot [8]. For functional pre- and postoperative evaluations, the American Orthopedic Foot and Ankle Score (AOFAS score), EQ-5D General Life Quality Rating, and Visual Analogue Scale (VAS) were used.

### 2.4. Statistical Analysis

Descriptive statistics of all data were calculated (average, standard deviation, median, minimum, maximum, interquartile range—IQR) and the normality hypothesis of quantitative variants was examined using the Shapiro–Wilk test. For comparisons between groups, independent t tests and Mann–Whitney U tests were employed. We estimated parameters using the generalized estimating equations method to compare the evaluation scores between the groups and using the most suitable model (Gamma with log link, ordinal probit model; post hoc: LSD). Fisher’s exact and Fisher–Freeman–Halton tests were used to examine relationships among categorical variants. The intraclass correlation coefficient was calculated to evaluate the reliability of radiological measurements. Statistical evaluations were performed using the SPSS program where *p* < 0.05 was considered significant. 

## 3. Results

The study sample was 75% men and 25% women, and the average age was 38.46 ± 14.62 (15–67), without a significant difference between groups (*p* = 0.560). However, the follow-up period was significantly longer in Group 2 than in Group 1 (*p* < 0.001). Other clinical features of the patients are provided in Table 1. Flow diagram of results can be seen in Figure 3.

Descriptive values and comparison results of HVA and IMA of left and right foot measurements are provided in Table 2. A significant result was defined as either a difference between the measurements of hva_left measured during the pre- and postoperative periods or changes between the groups between each period (*p* = 0.004). Based on the post hoc test results, the postoperative hva_left angle score was significantly higher in Group 2 than in Group 1 (*p* = 0.015). The hva_left score measured in Group 1 and 2 during the preoperative period was significantly higher than the score during the postoperative period (*p* < 0.001). Furthermore, the change in hva_left angle score in Group 1 was 159% greater than the change observed in Group 2 (*p* = 0.004).

Based on post hoc test results, the postoperative hva_right angle score was significantly higher in Group 2 than in Group 1 (*p* = 0.050). It was significantly higher during the preoperative period than during the postoperative period in both groups (*p* < 0.001). The change in hva_left angle score was 186% greater in Group 1 than in Group 2 (*p* = 0.023). The differences in the ima_right measurement were similar during the pre- and postoperative period in both groups (*p* = 0.619). The ima_left score was significantly higher during the preoperative period in both groups (*p* < 0.001). Similarly, the postoperative ima_right angle score was significantly higher in Group 2 than in Group 1 (*p* = 0.047), and that in Group 2 was significantly higher during the preoperative period than during the postoperative period (*p* < 0.001). Furthermore, the change in that group was 203% higher than that in Group 1 (*p* = 0.003). The intraclass correlation coefficients calculated to evaluate the reliability of the HVA–IMA and implication measurements were *r* = 0.817 (*p* = 0.005) and *r* = 0.678 (*p* = 0.040) in Group 1, in addition *r* = 0.775 (*p* = 0.002) and *r* = 0.761 (*p* = 0.003) in Group 2 respectively.

The descriptive statistics and AOFAS and VAS scores are provided in Table 3. AOFAS scores were significantly lower during the preoperative period in both groups (*p* < 0.001), and the change in AOFAS scores was 133% higher in Group 1 than in Group 2 (*p* < 0.001).

VAS scores were significantly higher in Group 2 than in Group 1 (*p* = 0.0001). The IMA VAS score was significantly higher during the preoperative period in both groups (*p* < 0.001), and the change in VAS score was 330% higher in Group 1 than in Group 2.

The descriptive statistics and comparative results of EQ-5D scores are provided in Table 4. The postoperative scores were significantly higher in Group 2 than in Group 1 (*p* = 0.0001), but overall preoperative scores were higher than postoperative scores in both groups (*p* < 0.001).

There was no statically significant difference of the EQ-5D Overall Quality of Life Scale between Group 1 and Group 2 (*p* > 0.05). However, the overall total scores were significantly lower during the preoperative period than during the postoperative period in both groups (*p* < 0.001). Furthermore, the change in total score was 122% higher in Group 1 than in Group 2 (*p* < 0.009). 

## 4. Discussion

A high correlation between clinical, radiological, and functional evaluations, as well as personal satisfaction are important indicators of successful surgical treatment. In a surgery, it is important to maintain the movements of the first MTP joint and load bearing function of the fore foot. Patients expect pain relief, an improvement in cosmetic appearance, and recovery in functions of the foot (in particular of the big toe). In this retrospective study, we compared the effectiveness of the two treatment modalities that have been widely used for the treatment of hallux valgus deformity—intramedullary self-locked plates and Herbert screws—and discussed the results based on healing time and return to daily life in the light of the pertinent literature.

Although many types of osteotomy have been defined in the medical literature, for advanced deformities in which the intermetatarsal angle is more deformed, more proximal corrective osteotomies are preferred [7,9,10,11]. 

In this study, HVA and IMA were used as radiological evaluation criteria. The short postoperative recovery periods of the Lindgren and Turan osteotomies were evaluated and had the most evident advantage [10]. However, it was not preferred in treatments of advanced hallux valgus cases as it was a distal osteotomy [12]. 

Coughlin and Jones examined 122 feet and showed that HVA recovered an average of 20 degrees, while IMA recovered nearly 9.1 degrees [13]. Tatar et al. reported that the mean HVA of the preoperative, 32.2°, was 14.4° postoperatively and IMA were reduced from 13.3° to 9.8° after treatment [14]. Ertürer et al. reported that the mean HVA of the preoperative, 27.1°, was 13.5° postoperatively and IMA were reduced from 22.0° to 8.4° after treatment and showed that these results were significant [15]. In our study, in agreement with previous studies, preoperative HVA and 1–2 IMA values were significantly lower than postoperative values. 

Esemenli et al. evaluated the patients using Bonney and Macnab criteria, and reported 80% very good, 16% good, and 4% bad results [11]. Yücel et al. reported an AOFAS of 39.2 ± 14.1 during the preoperative period and 87.1 ± 6.2 during the postoperative period [16]. In agreement with these reports, we found that AOFAS scores were significantly higher postoperatively in both groups.

In a previous study, 20% of post-distal osteotomies led to avascular necrosis of the metatarsal [17]. We did not observe any avascular necrosis in our patients. A total of five patients in Group 2 experienced complications including transfer metatarsalgia in three patients and wound hypoesthesia and delayed union in the other two patients.

Yücel et al. reported that VAS scores decreased markedly after surgical correction of the deformity and the same was true in our study [16]. In line with Şükür et al., we found that surgery improved quality of life, although this improvement was significantly higher in Group 1 than in Group 2 [18]. 

The limitation of this study is the short follow-up period. More research is needed in this area to gain sufficient knowledge on other outcomes such as long-term quality of life measures and patient satisfaction. Therefore, we believe that this study is meaningful with regard to the choice of stabilization method after distal metatarsal osteotomy.

## 5. Conclusions

For the surgical treatment of hallux valgus, an operation performed using self-locked plates is a good choice compared to an operation using a single screw in terms of functional, radiological, and pain scores, and provides rigid stability. The self-locked plates system in HV can be considered as a safer, easier, and effective surgical option in HV surgery.

## Figures and Tables

**Figure 1 medicina-55-00595-f001:**
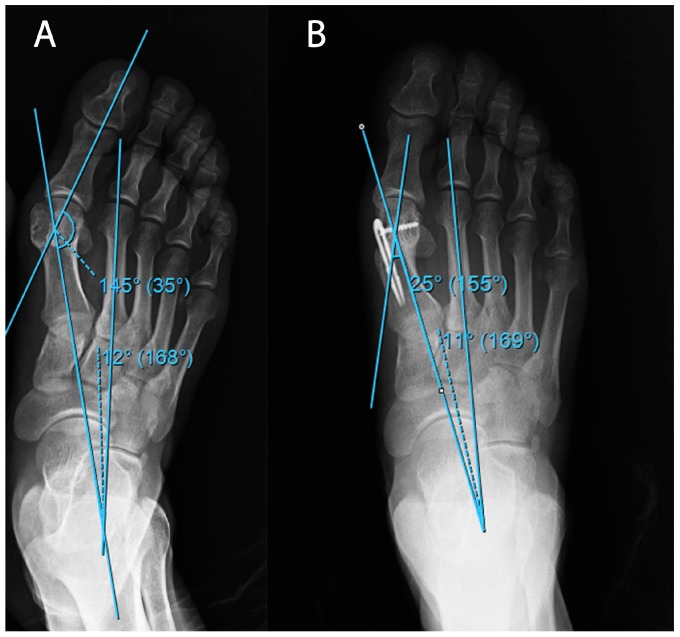
An example of (**A**) preoperative and (**B**) postoperative radiographic examinations of patients in Group 1.

**Figure 2 medicina-55-00595-f002:**
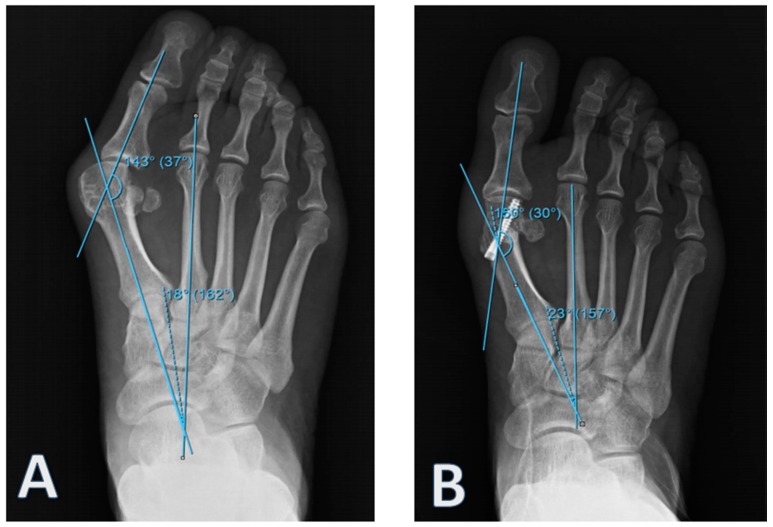
An example of (**A**) preoperative and (**B**) postoperative radiographic examinations of patients in Group 2.

**Figure 3 medicina-55-00595-f003:**
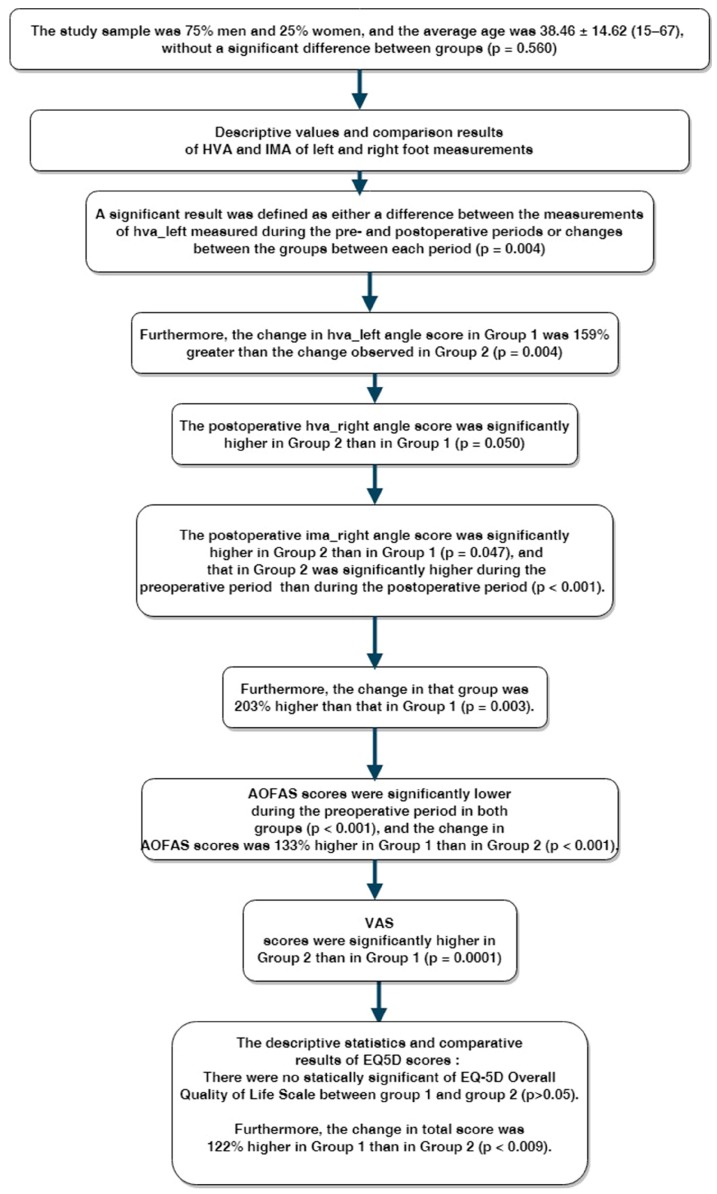
Flow diagram of results. HVA—hallux valgus angle; IMA—intermetatarsal angle; AOFAS—American Orthopedic Foot and Ankle Score; VAS—Visual Analogue Scale.

**Table 1 medicina-55-00595-t001:** Demographic information of patients.

		Group									*p*
		1			2			Total			
		*n*	R%	C%	*n*	R%	C%	*n*	R%	C%	
Sex	M	6	100.0	50.0	0	0.0	0.0	6	100.0	25.0	0.014
	F	6	33.3	50.0	12	66.7	100.0	18	100.0	75.0	
Side	Bilateral	6	40.0	50.0	9	60.0	75.0	15	100.0	62.5	0.376
	Right	4	80.0	33.3	1	20.0	8.3	5	100.0	20.8	
	Left	2	50.0	16.7	2	50.0	16.7	4	100.0	16.7	
Age *		36.67 ± 13.92 (18–57)			40.25 ± 15.69 (23–67)			38.46 ± 14.62 (18–67)			0.560
Follow-up period ^#^		5 (3–12)			14 (8–30)			12 (3–30)			<0.001

* Mean ± SD (standard deviation) (min–max); ^#^ Median (min–max); R%: Row %; C%: Column %; M: Male; F: Female.

**Table 2 medicina-55-00595-t002:** Radiographic results.

	Group	Period	*n*	Mean	SD	Median	Min–Max	IQR	*p*	OR for Group * Period (95% Wald CI)
**hva_left**	1	Preop	9	36.7	11.3	34.0	25–60	16	**0.004**	1.593 (1.160–2.190)
		Postop	8	11.0	6.2	10.0	5–25	5.5		
	2	Preop	11	37.3	8.3	38.0	27–52	16		
		Postop	11	17.8	6.6	18.0	7–32	9		
**hva_right**	1	Preop	10	39.5	8.0	38.5	30–56	12.3	**0.023**	1.856 (1.088–3.166)
		Postop	10	9.0	7.9	7.0	0–30	3.5		
	2	Preop	10	36.4	7.1	38.0	25–49	10		
		Postop	10	17.1	9.4	18.0	5–35	14.8		
**ima_left**	1	Preop	8	14.9	5.5	16.0	7–23	8.8	0.619	-
		Postop	8	7.9	3.8	7.5	3–13	6		
	2	Preop	11	16.6	5.1	17.0	10–24	10		
		Postop	11	7.7	3.0	8.0	4–12	6		
**ima_right**	1	Preop	10	13.0	3.1	13.5	7–17	4	**0.003**	2.028 (1.271–3.236)
		Postop	10	12.5	8.4	9.0	4–26	17.3		
	2	Preop	10	15.4	4.2	15.5	11–24	6		
		Postop	10	7.3	2.2	7.0	5–11	3		
**ICC (*p*)**	**hva**		0.817 * (0.005)		0.775 ^&^ (0.002)					
	**ima**		0.678 * (0.040)		0.761 ^&^ (0.003)					

SD: Standard deviation, Min: Minimum, Max: Maximum, IQR: Interquartile range, OR: Odds ratio, CI: Confidence interval, ICC: Intraclass correlation coefficient, * Group 1, ^&^ Group 2.

**Table 3 medicina-55-00595-t003:** Functional results.

	Group	Period	*n*	Mean	SD	Median	Min–Max	IQR	*p*	OR for Group * Period (95% Wald CI)
**AOFAS**	1	Preop	12	52.0	2.9	52.0	46–56	4.8	**<0.001**	1.333 (1.236–1.437)
		Postop	12	82.6	3.5	83.0	76–88	5.8		
	2	Preop	12	52.8	2.7	53.0	50–56	5.8		
		Postop	12	62.8	6.8	62.5	50–72	11.3		
**VAS**	1	Preop	12	6.8	0.8	7.0	6–8	1.8	**<0.001**	3.297 (2.605–4.173)
		Postop	12	1.6	0.8	1.0	1–3	1		
	2	Preop	12	6.0	1.0	6.0	5–8	2		
		Postop	12	4.6	0.8	5.0	3–6	1		

AOFAS: American Orthopedic Foot and Ankle Society, VAS: Visual Analogue Scale.

**Table 4 medicina-55-00595-t004:** EQ-5D Overall Quality of Life Scale.

	Group	Period	Median	Min	Max	IQR	*p*	OR for Group * Period (95% Wald CI)
a_movement	1	Preop	1.0	1	2	1	0.142	-
		Postop	1.0	1	2	0.8		
	2	Preop	1.0	1	2	1		
		Postop	2.0	1	2	1		
b_care	1	Preop	1.0	1	2	0	0.450	-
		Postop	1.0	1	1	0		
	2	Preop	1.0	1	2	0.8		
		Postop	1.0	1	2	0		
c_activity	1	Preop	2.0	1	2	1	0.716	-
		Postop	1.0	1	2	0		
	2	Preop	2.0	1	3	0		
		Postop	1.5	1	2	1		
d_pain	1	Preop	2.0	1	2	0.8	0.330	-
		Postop	1.0	1	2	0.8		
	2	Preop	2.0	1	3	0		
		Postop	2.0	1	3	1		
e_anxiety	1	Preop	2.0	1	3	1.5	0.003	1.500 (1.151–1.954)
		Postop	1.0	1	2	0		
	2	Preop	2.0	1	3	0		
		Postop	2.0	1	3	1		
total_pre	1	Preop	8.0	5	10	3.3	0.009	1.225 (1.052–1.27)
		Postop	6.0	5	7	0.8		
	2	Preop	8.5	7	10	1		
		Postop	8.0	5	10	2.8		

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
