# Peer review of "Comparison of Two Different Approaches to Treat a Hallux Valgus Deformity: Intramedullary Self-Locked Plates and Herbert Screws"

_medicina, 2019, doi:10.3390/medicina55090595_

Round 1

Reviewer 1 Report

INTRODUCTION

You should be to check the reference of the text (from line 35 to line 39,  there isn´t  any bibliography reference)

Although I know about extension enclosure needs, on this part of the paper, you shoul be to write about two kinds of variables to compare on your study (intramedullary self-locked plates and Herbert screws). In addition, I recommended to do reference to another articles, (for example https://www.ncbi.nlm.nih.gov/pubmed/30884845) to justify the three-dimensional deviation of both metatarsal and proximal phalanx bones deviation.

In this section and for the full text too, I recommended replace the writting "personal satisfaction" by " subjects satisfaction"

MATERIAL AND METHODS

There is no reference about technical characteristics related to X-Ray device, neither physics parameters of distance to focus, collimation, etc of x-ray shots and these are a very important things to take into acount.

Figure 1, A is distorted; please modify it

STATISTICAL ANALYSIS

It is lack of Intraclass correlation coefficient, Size effect and Minimal Detectable Change in order to ensure that radiological measurements are inside quality assessments values of validity and reliability.

RESULTS

I recommend to do a flow chart of partcipants to show more clarity on this

section.

Line 139-141: it is needed to re-write it,  to show non-statistical results of this section

CONCLUSIONS

Line 183.- I recommend replace the word "superior" and to write another expression

Author Response

Responce to Comments and Suggestions

Reviewer 1:

INTRODUCTION

You should be to check the reference of the text (from line 35 to line 39,  there isn´t  any bibliography reference)

Thank you for your valuable attention. The necessary changes are tried to be done in introduction sections and highlighted in the text.

Although I know about extension enclosure needs, on this part of the paper, you shoul be to write about two kinds of variables to compare on your study (intramedullary self-locked plates and Herbert screws). In addition, I recommended to do reference to another articles, (for example https://www.ncbi.nlm.nih.gov/pubmed/30884845) to justify the three-dimensional deviation of both metatarsal and proximal phalanx bones deviation.

Thank you for your comment and recommendation. The necessary changes are tried to be done in introduction sections and highlighted in the text.

In this section and for the full text too, I recommended replace the writting "personal satisfaction" by " subjects satisfaction"

Thank you for your advice. ‘Personal’ word replace with ‘subjects’

MATERIAL AND METHODS

There is no reference about technical characteristics related to X-Ray device, neither physics parameters of distance to focus, collimation, etc of x-ray shots and these are a very important things to take into acount.

Thank you for your valuable attention and comment. The necessary changes are tried to be done in methods sections and highlighted in the text.

Figure 1, A is distorted; please modify it.

Thank you for your valuable attention. We modified the figure 1A.

STATISTICAL ANALYSIS

It is lack of Intraclass correlation coefficient, Size effect and Minimal Detectable Change in order to ensure that radiological measurements are inside quality assessments values of validity and reliability.

Thank you for your valuable attention and comment. The necessary changes are tried to be done in methods sections and highlighted in the text.

RESULTS

I recommend to do a flow chart of partcipants to show more clarity on this

section.

Thank you for your valuable comment. We added a flow chart of results in result section.

Line 139-141: it is needed to re-write it,  to show non-statistical results of this section

Thank you for your valuable attention and comment. The necessary changes are tried to be done in results sections and highlighted in the text.

CONCLUSIONS

Line 183.- I recommend replace the word "superior" and to write another expression

Thank you for your valuable comment. We changed  and used ‘good choice’ instead of ‘superior’.

Reviewer 2 Report

First of all, I would like to congratulate the authors for the presented manuscript.

Here are my observations and questions after reading and reviewing in deep the manuscript:

In the Materials and Methods section, you explain that the patients were classified with high to moderate Hallux Valgus, have you used any validated Hallux Valgus classification scale?

One of the selection criteria of his study was to include participants over 18 years old; in the Results section, the age range of the participants was 15 to 67 years. There is some error between the exposed data.

It is expected that the changes described will be made.

Author Response

Reviewer 2:

First of all, I would like to congratulate the authors for the presented manuscript.

Here are my observations and questions after reading and reviewing in deep the manuscript:

In the Materials and Methods section, you explain that the patients were classified with high to moderate Hallux Valgus, have you used any validated Hallux Valgus classification scale?

Thank you for your valuable comment. The necessary changes are tried to be done in method section and highlighted in the text.

One of the selection criteria of his study was to include participants over 18 years old; in the Results section, the age range of the participants was 15 to 67 years. There is some error between the exposed data.

Thank you for your valuable attention and comment. The necessary changes are tried to be done in result section and highlighted in the text.

It is expected that the changes described will be made.

Round 2

Reviewer 1 Report

1.-Please, you have to identify the lines-number where you have made the improvements on your text when you submit any corrections to reviewers.

2.-I have not find the improvements to this question: "There is no reference about technical characteristics related to X-Ray device, neither physics parameters of distance to focus, collimation, etc of x-ray shots and these are a very important things to take into acount".

3.-You have to show ICC results inside a table, not only in results text. Please, insert these results in one of your tables.

4.- I recommended that you place your flow chart begining on "results section", not at the end.

Author Response

1. Please, you have to identify the lines-number where you have made the improvements on your text when you submit any corrections to reviewers.

Thank you for your advice. We made some revisions in the text as;
Line 6 and 10: Newly added author’s name and instituion was written.
Line 45: “subject” was added.
Lines 48 and 49: “the two treatment options (Intramedullary Self-Locked Plates and Herbert Screws) of hallux valgus patients after distal metatarsal osteotomy.” sentence is revised.
Line 53: The number of ethics was changed (the mistake is corrected).
Line 61: The reference 7 is added.
Line 63: The figure 1 is changed.
Line 83-85: The reference 8 was added and the radiographic measurements were explained.
Line 96: “Intraclass correlation coefficient was calculated” and added to the text.
Line 101: The figure 3 was replaced to the begginning of the results section.
Line 126-128: Table 2 was revised and ICC results were added to the table 2.
Line 139: “The intraclass correlation coefficients calculated” and the results were given in this section.
Line 165: The EQ-5D results were added.
Line 210: superior was changed as “good choice”
Line 231-233: references 7 and 8 were added.

2. I have not find the improvements to this question: "There is no reference about technical characteristics related to X-Ray device, neither physics parameters of distance to focus, collimation, etc of x-ray shots and these are a very important things to take into acount".

Thank you for your remind. The radiographic measurements were tried to be explained as: “During radiographic imaging, the tube is placed 100 cm away from the foot and focusing on the midtarsal joint. The tube is than rotated at a 15° angle to the ankle relative to the plantar face of the foot [8]”. in the text and a new reference was added.

3. You have to show ICC results inside a table, not only in results text. Please, insert these results in one of your tables.

Thank you for your advice. The results of ICC were added to the table 2.

4. I recommended that you place your flow chart begining on "results section", not at the end.

Thank you for your recommendation. The chart was replaced to the begginning of results section.
